# Rational Design and Characterisation of Novel Mono- and Bimetallic Antibacterial Linde Type A Zeolite Materials

**DOI:** 10.3390/jfb13020073

**Published:** 2022-06-02

**Authors:** Emmanuel Oheix, Chloé Reicher, Habiba Nouali, Laure Michelin, Ludovic Josien, T. Jean Daou, Laurent Pieuchot

**Affiliations:** 1Institut de Science des Matériaux de Mulhouse (IS2M), Université de Haute Alsace (UHA), CNRS, UMR 7361, 3 bis rue Alfred Werner, F-68093 Mulhouse, France; emmanuel.oheix@uha.fr (E.O.); chloe.reicher@uha.fr (C.R.); habiba.nouali@uha.fr (H.N.); laure.michelin@uha.fr (L.M.); ludovic.josien@uha.fr (L.J.); 2Université de Strasbourg (UniStra), F-67000 Strasbourg, France; 3Aptar CSP Technology, 9 rue du Sandholz, F-67110 Niederbronn les Bains, France

**Keywords:** zeolite, metal-ions, antibacterial properties, metal loading, quantification, minimum inhibitory concentration, agar dilution, optical density, *Escherichia coli*

## Abstract

The development of antimicrobial devices and surfaces requires the setup of suitable materials, able to store and release active principles. In this context, zeolites, which are microporous aluminosilicate minerals, hold great promise, since they are able to serve as a reservoir for metal-ions with antimicrobial properties. Here, we report on the preparation of Linde Type A zeolites, partially exchanged with combinations of metal-ions (Ag^+^, Cu^2+^, Zn^2+^) at different loadings (0.1–11.9 wt.%). We combine X-ray fluorescence, scanning electron microscopy, energy-dispersive X-ray spectroscopy, and X-ray diffraction to monitor the metal-ion contents, distribution, and conservation of the zeolite structure after exchange. Then, we evaluate their antimicrobial activity, using agar dilution and optical-density monitoring of *Escherichia coli* cultures. The results indicate that silver-loaded materials are at least 70-fold more active than the copper-, zinc-, and non-exchanged ones. Moreover, zeolites loaded with lower Ag^+^ concentrations remain active down to 0.1 wt.%, and their activities are directly proportional to the total Ag content. Sequential exchanges with two metal ions (Ag^+^ and either Cu^2+^, Zn^2+^) display synergetic or antagonist effects, depending on the quantity of the second metal. Altogether, this work shows that, by combining analytical and quantitative methods, it is possible to fine-tune the composition of bi-metal-exchanged zeolites, in order to maximise their antimicrobial potential, opening new ways for the development of next-generation composite zeolite-containing antimicrobial materials, with potential applications for the design of dental or bone implants, as well as biomedical devices and pharmaceutical products.

## 1. Introduction

Some transition metal ions, such as Mn^2+^/Mn^3+^, Fe^2+^/Fe^3+^, Co^2+^/Co^3+^, Cu^+^/Cu^2+^, and Zn^2+^, are naturally present in living cells, where they play crucial biological roles (electron transfer, catalytic or structural roles) [1,2]. Yet, these essential metal ions can disrupt the cellular machinery, when present in excess. Besides, some exogenous metal ions, such as Ag^+^ and Hg^2+^, display acute cellular toxicity [3,4]. Lethal concentration of metal ions (essential and non-essential) is associated with multiple dysfunctions resulting from oxidative stress, metalloprotein dysfunction, and cell membrane disruption. Their combinations act in such a way that it is often not possible to pinpoint one particular cell-death mechanism [5]. The membrane transporters and metal regulation systems vary depending on the organism, and, therefore, the critical level of a given metal ion is different for bacteria and mammals. Some metals, such as Cu and Ag, have a long-standing history as antimicrobial agents [6,7], and they are, still, broadly used nowadays in medical devices [8] (for example as catheters, bandages, wound dressing) and food packaging [9]. In the context of antimicrobial resistance, metal-ion salts, metallic surfaces, and metal nanoparticles, currently, are regaining interest as antibacterial compounds. Current developments aim for carrier materials (organic [10] or inorganic [11,12,13,14,15]), which serve as a reservoir, able to release metal ions (or nanoparticles) at sufficient rates for the antimicrobial activity to be retained but moderate enough to preserve a long-standing antimicrobial activity. For example, a food-storage container made of polymer, embedded with a low amount of Ag nanoparticles (~0.001 wt.% Ag), was found ineffective in inhibiting bacterial growth, presumably resulting from the low Ag^+^ release [16,17].

Zeolites are microporous crystalline minerals, made of well-organised assemblies of SiO_4_ and AlO_4_^−^ tetrahedra. The negative charges are compensated by cations (usually Na^+^), which are bound to the zeolite framework by weak interactions and are, therefore, exchangeable. Ag, Cu, and Zn-exchanged zeolites [18,19,20,21,22,23,24,25,26,27,28,29,30,31,32,33,34,35,36,37,38,39,40,41,42,43,44], as well as zeolite-based composites [45,46,47,48,49,50,51,52,53] and coatings [54,55,56,57,58,59,60,61,62], are well-studied antimicrobial materials (for review, see [63,64]), which have been commercialised. The exchanged-zeolite frameworks release transition metal-ions, by cationic exchanges, and their antimicrobial efficiency are, therefore, related to the salt content of the solution or deposit. Despite the large number of studies published, there remain large discrepancies in the testing conditions (quantification methods, cell counts, incubation times, bacteria types) and some important questions remain open to date, such as: (i) the impact of the zeolite structure and the Si/Al ratio, (ii) the metal ion with the highest effectiveness/cost ratio, (iii) the optimal metal loading (high metal loading favours the formation of metal aggregates that are, potentially, inactive), and (iv) the synergistic effect of different metal ions included in one material. For the latter, I. C. Neves et al. used an adaptation of the agar dilution method to compare the growth inhibition of *E. coli* and *S. cerevisae* in the presence of FAU-type zeolite exchanged with one or two of the following salt solutions: Ag(NO_3_) (0.01 M), Cu(NO_3_)_2_ (0.01 M), and/or Zn(NO_3_)_2_ (0.05 M) [65]. From this study, it was concluded that Ag-exchanged zeolites are the most effective at inhibiting bacteria and yeast growth, and that synergistic effects are obtained upon addition of Cu^2+^ and Zn^2+^, with the combination Ag_0.01_/Zn_0.05_ giving the best results. However, the conclusions drawn from this study are limited, considering the lack of metal quantification in the exchanged materials and the divergence in metal salt concentration for the exchange process. In solution, synergistic antibacterial effects have been studied for different combinations of salts Ag^+^/M^2+^ (with M = Ni, Cu, Zn), but their ranking varied with the organism studied [66]. Therefore, we argue that the order of synergistic antibacterial effects reported for metal-exchanged zeolites could benefit from complementary studies.

Our group previously reported on the antimicrobial activity of a FAU-type zeolite membrane synthesised on the surface of an alumina support [67]. In this study, we selected a common zeolite with low Si/Al ratio (=1) associated with a high content of negatively charged aluminates, the Linde type A (LTA) (Figure 1), thus allowing for metal-exchange at a high rate, whilst limiting the formation of undesired metal aggregates. The antimicrobial properties of Ag-exchanged LTA have been, previously, tested at different loadings (2.5–42.7 wt.% Ag^+^) [18,19,20,21,22,23], but the testing methods and conditions largely differ, thus precluding any comparisons. Moreover, no studies were performed at low Ag loading (≤2.5 wt.% Ag^+^). From an industrial standpoint, antimicrobial materials should be cost-effective and uncoloured, two criteria better met by zeolites with low Ag-loading. Therefore, our primary objective is to study the correlation between Ag^+^ loading and the antimicrobial activity of the Ag-exchanged zeolite as well as to determine whether the antimicrobial activity is retained at low Ag-loading. As secondary objectives, we aim to test and compare different methodologies to evaluate the antimicrobial activity. Gram-negative bacterium *Escherichia coli* (*E. coli*) is used to assess the antimicrobial activity, by two distinct methodologies: agar dilution and optical density (OD) monitoring at 595 nm. The first method is the gold standard, whereas the second method is homemade and adapted for measurements on a multiplate reader, for fast screening. In our setup, similar results are obtained for the OD monitoring and agar dilution methods, thus confirming the efficacy of the former. Finally, we aim to evaluate the synergistic antibacterial effects for bimetallic Ag/Cu and Ag/Zn-exchanged LTA zeolites as well as to compare the results obtained with previous reports.

Therefore, this study reports on the preparation of monometallic and bimetallic LTA-zeolites, by cationic exchange of parent Na-LTA zeolite, with Ag^+^, Cu^2+^, and/or Zn^2+^ solutions, at different concentrations. The metal content of the exchanged zeolites was measured by X-ray fluorescence, its distribution was assessed by energy-dispersive X-ray spectroscopy, and the integrity of the zeolite structure was controlled by X-ray diffraction. The ability of the monometallic and bimetallic zeolites to inhibit *E. coli* growth was then assessed by agar dilution and OD monitoring. Our results indicate that the Ag-exchanged LTA series display the highest antibacterial activity among the monometallic LTA tested, and that their activity can be further improved upon addition of a second metal ion (either Cu or Zn) at low loading. These formulations will be further used for the development of composite materials, with broad applications for the development of medical devices and packaging.

## 2. Materials and Methods

### 2.1. Generalities

All chemicals were of analytical grade and used as purchased. Linde type A zeolite was (Aptar CSP Technologies, Inc., Niederbronn les bains, France) stored in a desiccator over a saturated aqueous solution of NH_4_Cl, to ensure a constant water content (80% relative humidity). For the cationic exchanges, the following metal salts were used: Ag(NO_3_) 99.9 + % (Alfa-Aesar, Thermo Fisher Scientific, Waltham, MA, USA), Zn(NO_3_)_2_ 99 + % (Sigma-Aldrich, Merck KGaA, Darmstadt, Germany), and Cu(NO_3_)_2_ 99 + % (Alfa-Aesar, Thermo Fisher Scientific, Waltham, MA, USA). Thermogravimetric (TG) measurements were used, to determine the water content of the zeolite. The zeolite material (20–35 mg) was inserted in a 70 µL alumina crucible and analysed on a STARe apparatus (Mettler Toledo, Greifensee, Switzerland), using a 5 °C/min gradient from 30 °C to 800 °C, under air flow. The water content was determined from the weight loss over this temperature range and is displayed in Appendix A. Elemental contents of the zeolite samples were measured by X-ray fluorescence (XRF) with a Zetium (4 kW) spectrometer (Malvern PANalytical, Almelo, The Netherlands). A 13 mm pellet was prepared, by compressing 200 mg of finely ground zeolite material with 100 mg boric acid with 5 tons pressure. The weight percentage (wt.%) of transition metals (Ag, Cu, Zn) non-corrected for hydration rates are displayed in Appendix A. For scanning electron microscopy (SEM) and energy-dispersive X-ray spectroscopy (EDX), the zeolite powder was coated with carbon to ensure good conductivity. SEM was performed on a High Resolution JSM-7900F (JEOL, Tokyo, Japan), using an accelerating voltage of 5 kV and a 10 mm working distance. EDX was performed on a QUANTAX system (Bruker Nano GmbH, Berlin, Germany), equipped with two XFlash6-30 detectors operating at 15 kV SEM accelerating voltage. X-ray diffraction (XRD) patterns were collected on a MPD X’Pert Pro diffractometer (Malvern PANalytical, Almelo, The Netherlands), operating with Cu K_α_ radiation (K_α_ = 0.15418 nm) equipped with a PIXcel real-time multiple strip detector (active length = 3.347° 2θ). The powder XRD patterns were collected at room temperature, in the range 3° < 2θ < 70°, with step = 0.013° 2θ and time/step = 220 s. The phase identification has been realised with Highscore Plus software (PANanalytical) and the PDF-4+2022 database from the International Centre for Diffraction Data (ICDD, Newtown Square, PA, USA). Antibacterial test was performed using an *E. coli* DH5 alpha strain (*Escherichia coli* BL21 strain, ATCC ref. BAA-1025) resistant to ampicillin. It was grown in a lysogeny broth (LB) made of peptone (10 g/L), yeast extract (5 g/L), and NaCl (10 g/L), and containing ampicillin (50 µg/mL).

### 2.2. Preparation of the Metal-Containing LTA Zeolites

The metal-containing zeolites were prepared, by the method previously reported [65]. In total, 50 mL of solutions were added to a polypropylene (PP) tube containing LTA zeolites (3.097 g hydrated zeolite accounting for 2.5 g anhydrous, 1.428 mmol), containing different concentrations of metal cations as a nitrate salt (Ag(NO_3_): 0.001–0.005–0.01–0.02–0.05–0.1 M, Cu(NO_3_)_2_, and Zn(NO_3_)_2_: 0.005–0.01–0.025–0.05–0.1–0.5 M). A magnetic stirrer bar was added, and the suspension was stirred at room temperature for 24 h, at which point, the supernatant solution was removed. The exchanged zeolite was washed with deionised water (3 × 50 mL), via successive cycles of dispersion and centrifugation, and, then, it was dried in the oven for 15 h at 80 °C. For the bimetallic solutions, the zeolite was successively treated with the solutions of the two metal salts (0.01 or 0.05 M), in the order mentioned (for example: Zn/Ag = Zn(NO_3_)_2_ first, AgNO_3_ second), and washed with water (3 × 50 mL) after each step. Silver-exchanged zeolite were protected from light (to avoid a reduction in Ag^+^ cations) during storage. Each cationic exchange was performed in duplicate, to check the reproducibility.

The metal content of the exchanged-LTA samples was determined from X-ray fluorescence (XRF) measurements. This technique allows for a semi-quantitative determination of the elemental composition, however, the composition measured does not account for the contribution of low-molecular weight elements (notably, H and O). Therefore, the humidity of the samples was determined from thermogravimetric (TG) measurements, and the metal content measured by XRF was, subsequently, corrected The corrected results are displayed in Table 1 and Table 2 and were used to calculate the molecular formula in the following manner: (1) the silicium, aluminium, and oxygen content are, respectively, fixed at 12, 12, and 48, per unit cell, and (2) the molar content of the other elements was calculated by proportionality from the aluminium content. The exchange rate (ER) was calculated from the following equation:ER (M) = 100 × δ_M_ × n_M_/(2n_Cu_ + 2n_Zn_ + n_Ag_ + n_Na_ + n_K_)(1)

Here, δ_M_ represents the charge/valence (1 for Ag^+^, 2 for Cu^2+^ and Zn^2+^) and n_M_ the molarity in the chemical formula, for a metal M.

### 2.3. AGAR Dilution Method

A pre-culture of *E. coli* was, first, prepared by incubation at 37 °C overnight. A fresh culture (50 mL) was, then, started by diluting 10-fold the pre-culture in lysogeny broth (LB), followed by 3 h incubation at 37 °C. At this point, the OD was measured and the bacteria concentration was determined, considering that 1 OD unit = 5.10^8^ CFU/mL (based on repeated calibration by AGAR dilution). Besides, suitable zeolite suspensions in LB were prepared in 15 mL PP tubes, using vortex agitation and ultrawave sonication for 3 min. The amount of zeolite used was either weighted on a precision balance (such as m ≥ 1 mg) or taken from a well-shaken stock suspension of zeolite in LB (at 1 or 5 mg/mL). The reaction was initiated, by adding the bacteria suspension to the zeolite suspension, to obtain a reaction mix with V_total_ = 5.0 mL and [bacteria] = 2.10^8^ CFU/mL. This mixture was incubated for 2 h at 37 °C under rotatory agitation (160 rpm). At this point, each mixture was diluted 50,000-fold in LB, by three successive steps (100-fold, 50-fold, and, then, 10-fold), and 50 µL of each diluted mixture was spread on an AGAR plate. The plates were allowed to dry and were incubated overnight at 37 °C. The number of colony-forming units (CFU) was, then, counted manually and expressed as a percentage of viable bacteria (%VB), using the following equation:%VB = 100 × BP_sample_/BP_control_(2)

Here, BP_sample_ and BP_control_ represent the bacteria population for the sample and the control, respectively (NB: the control represents the culture without zeolite from the same series).

For each sample, the antimicrobial tests were performed, at six different zeolite concentrations, in duplicate. The %VB was plotted versus the logarithm of zeolite concentration (mg/mL) or the total Ag content (M), with the latter representing the concentration of the solution upon total Ag release.

### 2.4. Optical Density Measurements (Microplate Reader)

On a 96-well plate, 100 µL of LB was added in each well. Then, 100 µL of each zeolite stock suspension (6, 3, or 1.5 mg/mL) in LB was added to the first well of each line, and the zeolites were, then, diluted in series over their respective lines. At this point, the reaction was initiated, by adding 50 µL of the bacteria stock suspension in each well. Overall, each well contains 150 µL of a suspension, in which the bacteria were diluted 3-fold compared to the stock suspension, whilst zeolites were diluted 3-fold in the first well of each line, 6-fold in the second, 12-fold in the third, etc. The 96-well plate was then inserted in the microplate reader, pre-equilibrated at 37 °C, and the OD of each well was monitored over 2 h (Δt = 5 min). The OD slope values represent the derivatives of the curves, OD × 1000 = f (incubation time), with the time in min, for each zeolite material and concentration. For each material, the OD derivative accounting for the bacterial growth rate is measured at 8 or 12 different concentrations, and the OD slope is plotted versus the logarithm of zeolite concentrations.

## 3. Results and Discussion

### 3.1. Preparation of Metal-Exchanged Zeolites

The first step involves the preparation of partially cation-exchanged LTA zeolites, from the sodium-containing LTA (Na-LTA) starting material (Figure 1). For this, Na^+^ cations are, partially, exchanged by Ag^+^, Cu^2+^, and Zn^2+^, in one step for monometallic exchanged samples and in two successive steps for bimetallic exchanged samples. The metal-salt concentrations and the resulting metal contents for the selected samples are detailed in Table 1 and Table 2.

### 3.2. Determination of the Metal Content

The metal content of the exchanged-LTA samples determined from X-ray fluorescence (XRF) measurements are outlined in Table 1 and Table 2. For a given metal ion, the metal content and the exchange rate increase, proportionally, with the metal-ion concentration of the solution used for the exchange. However, the calculated molecular formula indicates that the total amount of positive charges per unit cell is higher than the amount of negative charges generated by the presence of tetracoordinated aluminium (AlO_4_^−^ aluminate anions). Therefore, cations in excess are either present as extra-framework species or localised at the external surface, where they, probably, compensate external negative charges associated to structural defects (Si-O^−^ or Al-O^−^). The metal content, being lower for the divalent cations (Cu^2+^, Zn^2+^) compared to the monovalent one (Ag^+^), is consistent with the fact that each divalent cation can replace two sodium cations (versus one for the monovalent Ag^+^). For the bimetallic samples, it appears, in most cases, that the proportion for a metal ion is higher, when it is introduced in the second exchange rather than the first.

### 3.3. Metal Ion Distribution and Structural Content

After measuring the total metal content of the exchanged zeolite, it is crucial to determine the distribution of metals within the zeolite materials. The metal ions are expected to be present in the micropores, but a portion might, also, be present on the external surface (in form of cations, metallic or hydroxide species) of the microcrystals, as previously reported [29,69]. In order to estimate the metal-ion distribution, some representative samples (down to 0.9–1.2 wt.% metal content) were analysed, by scanning electron microscopy (SEM) coupled with energy-dispersive X-ray (EDX) spectroscopy (Figure 2 and Appendix A). For the LTA sample containing 5.3 wt.% Ag, the EDX spectra overlap with the SEM images, indicating that the Ag is homogeneously spread over the crystalline material, and little cluster or agglomerate can be distinguished (Figure 2a,d); this is, also, the case of the highly loaded silver Ag-LTA-7 sample (11.9 wt.%) (Appendix A).

In contrast, the EDX spectra indicates that the Cu^2+^ ions are not homogeneously spread in the LTA containing 4.65 wt.% Cu (Figure 2b,e). Indeed, a portion of the crystals seem to be copper-enriched, from the EDX spectra and the SEM images displaying a thin fibrous deposit over the surface of these very same crystals. In the case of LTA containing 4.55 wt.% Zn, some Zn-enriched crystals (according to EDX) covered with fibres (based on SEM) are also observed, but their occurrences are much lower compared to the Cu-exchanged samples (Figure 2c,f). For Cu- and Zn-exchanged materials, the proportion of crystals with a fibrous deposit increases with the metal content. The presence of fibres would be consistent with the formation of a metal hydroxide layer on the material surface, but this remains a minor phenomenon at low metal content, and the biggest part of the metal is well dispersed over the material. A similar trend is obtained for the bimetallic samples (not reported), that is, the Ag atoms are uniformly spread over the crystalline material, whereas a low portion of Cu and Zn are present as agglomerates (associated with the presence of fibres) at the external surface of zeolite particles.

The samples were analysed by XRD, in order to check the impact of the metal-exchange process over the zeolite structure. The XRD pattern of the LTA sample containing 5.3 wt.% Ag displays a large increase in the intensity of the peak at 2θ = 14.4 ° and, conversely, a decrease of the peak’s intensities at 2θ = 17.7, 20.5 and 22.9 ° compared to the parent Na-LTA zeolite sample (Figure 3a,b). In the non-exchanged Na-LTA sample, the peak at 2θ = 14.4 ° is not observed, but a search on the International Center for Diffraction Database (ICDD) indicates that this LTA-associated diffraction peak usually displays a low relative intensity (see ICDD card 04-016-9920, for an example) [70]. A silver-exchanged zeolite, with exalted intensity of the peak at 2θ = 14.4 °, has been previously reported (ICDD card 04-009-1954) [71]. Further increases of the Ag content (11.9 wt.%) do not significantly affect the XRD pattern (Appendix A), but changes in peak intensities are observed, which are consistent with cationic-exchange processes [72]. Overall, the data indicate that high crystalline contents are retained, for all Ag-exchanged LTA prepared.

In the case of LTA zeolites containing 4.7 wt.% Cu or 4.6 wt.% Zn, the same XRD pattern is observed, but the peak intensities decrease slightly, and the peaks are slightly shifted towards high 2θ values (Figure 3c,d). The XRD pattern of the Zn-exchanged sample also display a peak at 14.4° and a decrease in intensity for the peak at 20.5°, whereas this is not the case for the Cu-exchanged sample. At higher copper loading, a significant broadening of all peaks is observed, indicating that the crystallinity of the material decreases (data not shown). In contrast, the XRD peaks of LTA samples with Zn loading higher than 4.6 wt.% are, further, shifted towards high 2θ values, with the shift extent increasing with 2θ. It remains unclear whether this shift is associated with the structural damage of the zeolite framework. Therefore, the samples with low Cu and Zn content are preferred.

### 3.4. Antibacterial Measurements

The ability of the metal-exchanged LTA to inhibit the growth of *E. coli* was measured by two distinct techniques: agar dilution and optical-density monitoring. In all cases, the foreseen process involves cationic exchanges between the zeolite and the NaCl-rich culture media (liquid-phase or agar gel), followed by metal ions diffusion in the liquid phase, where they exert their antimicrobial activity. For starters, the antimicrobial activities of LTA, exchanged with either Ag, Cu, or Zn, are compared to that of the non-exchanged LTA material. For comparison purposes, the samples contain similar metal loadings (Ag: 5.3 wt.%, Cu: 4.7 wt.%, Zn: 4.6 wt.%), corresponding to the samples with higher Cu and Zn loading, without alteration of the LTA framework (see previous section). The results of agar dilution for the different materials are shown in Figure 4a. For the Ag-exchanged sample, the %VB curve undergoes a three-stage slope with: (i) a high plateau (near 100%) at low zeolite concentration, (ii) a sharp decrease at intermediary concentrations, and (iii) a low plateau at 0% for high concentrations. From this curve, it is possible to approximate the minimum inhibitory concentration (MIC) to 0.15 mg/mL and the minimum inhibitory concentration required to inhibit the growth of 50% of organisms (MIC50) to 0.05 mg/mL.

For the Cu-exchanged zeolite, a similar slope is observed, and it is possible to determine the MIC50 (= 3.5 mg/mL), but the 0%VB value is not reached in the concentration domain studied (0.2–10 mg/mL). For the Zn-exchanged and non-exchanged LTA samples, no significant decrease in the %VB is observed, in the concentration range studied. Overall, these measurements are consistent with the antimicrobial efficiency of LTA exchanged and non-exchanged samples studied, following the trend Ag-LTA > Cu-LTA > Zn-LTA and Na-LTA, with the higher activity of Ag-containing samples being consistent with previous measurements in solutions and on biofilms [3,66]. By comparison, the MIC50 for Cu-LTA is 70-fold higher than that measured for Ag-LTA, which is, also, consistent with the MIC measured for Ag^+^ and Cu^2+^ in solution (60 and 4000 µM, respectively). Despite the intermediary MIC values measured for Zn^2+^ in solution (2000 µM), no *E. coli* growth inhibition was observed, for up to 10 mg/mL (representing potentially 7000 µM of Zn^2+^, to be released in solution). This suggests that a low portion of Zn^2+^ is re-exchanged in the solution, or that it is released in a less cytotoxic form, and further studies would be required to understand this observation.

For the second method, the metal-exchanged LTA (Ag: 5.3 wt.%, Cu: 4.7 wt.%, Zn: 4.6 wt.%) and non-exchanged zeolites are, also, incubated in the *E. coli* culture (2.10^8^ CFU/mL), while the OD of the suspension is monitored. Considering that the zeolite OD at 595 nm is constant, the OD increase represents the bacterial growth. Under these conditions (2.10^8^ CFU/mL), the *E. coli* growth is linear over the initial 90 min and reaches a plateau afterwards. As the concentration of antimicrobial material increases, the growth rate concomitantly decreases, to finally reach a standstill, at which point the OD remains constant. In the case of Ag-exchanged zeolite, the plot OD slope = f (log[zeolite]) resembles an inhibition curve, from which the MIC (0.25 mg/mL) and MIC50 values (0.05 mg/mL) are extracted (Figure 4b). These values are similar to that measured by Agar dilution (considering the concentration step), and this indicates that the OD monitoring assay proposed gives reliable results, in this case. For Cu-, Zn-, and Na-LTA, the OD slopes remain constant, indicating that there is no inhibition of *E. coli* growth in the concentration range studied (0.2–10 mg/mL). On a side note, the optical density measurements at high zeolite concentrations (C > 1 mg/mL) lead to large OD variations over time, and the slopes measured are not reliable. This might arise from non-negligible light scattering by the microparticles, above this threshold. Therefore, the range of this technique is limited to low particle concentrations.

Considering the results obtained, the focus was set on Ag-exchanged zeolites and the LTA samples with different Ag-loadings (0.1 wt.%, 0.6 wt.%, 1.2 wt.%, 2.7 wt.%, 5.3 wt.%, 6.1 wt.%, and 11.9 wt.%), which were analysed by the agar-dilution method, at six different zeolite concentrations. The results for the different exchanged zeolites are expressed as %VB plotted versus the logarithm of zeolite concentration (Figure 5a) or total Ag content (Figure 5b). In all cases, the percentage of viable bacteria (%VB) undergoes a three-stage slope, as previously described. As expected, the plot versus zeolite concentration indicates that zeolites with higher metal content inhibit bacterial growth at lower concentrations. However, the plot versus Ag content indicates that the antibacterial activity is proportional to the total Ag content (Figure 5b). Therefore, the material retains its antimicrobial activity at low metal loading (down to 0.1 wt.% Ag), and, presumably, the quantity of Ag^+^ ions released in the solution is, largely, independent of the amount of zeolite present under these conditions. From this plot, it appears that total inhibition is reached for a total Ag content of 10^−4.1^ M = 79 µM in the culture media, a value similar to the reported MIC value of Ag^+^ ions towards *E. coli* determined in solutions (60 µM) [66].

To consolidate this result, each Ag-exchanged LTA material (0.1, 0.6, 1.2, 2.7, 5.3, 6.1 and 11.9) was analysed by OD monitoring at eight different concentrations, in order to plot inhibition curves. For each material, the bacterial-growth rate is plotted versus the logarithm of zeolite concentration (Figure 5c) or the logarithm of total Ag content (Figure 5d). In both cases, it appears that the inhibition curves obtained from OD measurements and the agar dilution display the same trend and similar inhibition concentrations (MIC = 10^−4.25^ M = 56 μM total Ag content for OD measurements). Therefore, the OD method can represent a suitable alternative to the agar dilution, providing that bacterial growth inhibition is associated with bactericide, rather than bacteriostatic, effects. More importantly, the standard deviations calculated from the experiment repetitions are lower for the OD measurements than for the agar dilution.

Then, the antimicrobial efficiencies of the bimetallic-exchanged LTA were evaluated by the same methods and compared to the monometallic Ag-LTA-3 (1.2 wt.% Ag). From the inhibition curves for both agar dilution and OD measurements (Figure 6a,b), it appears that AgCu, CuAg, AgZn, and ZnAg-LTA-1 all inhibit *E. coli* growth at lower concentrations, compared to the monometallic Ag-LTA-3 sample. In contrast, the samples AgZn_0.05_ and Zn_0.05_Ag-LTA-2 both display antibacterial activity lower, compared to monometallic Ag-LTA-3 samples. In all cases, the order of metal exchange is shown to have little-to-no impact on the antibacterial activity (after correction for the Ag^+^ content). An alternative display is proposed, in which the antimicrobial activity measured at a single concentration (0.25 mg/mL zeolite) is represented (Figure 6c,d), and the trend of activity is globally identical. In this case, the concentration is not corrected for the metal content, and one should take care not to overinterpret the difference in activity between CuAg/AgCu (~1.2 wt.% Ag) and ZnAg/AgZn (~1.4–1.5 wt.% Ag). In summary, the addition, of approximately 0.7–1.4 wt.% Cu^2+^ or Zn^2+^ to the LTA material, containing 1.2–1.5 wt.% Ag, improve the overall antibacterial effects, whereas the addition of 3.9 wt.% Zn^2+^ is shown to have detrimental effects. This latter trend is opposite to that obtained by Neves et al. for bimetallic-exchanged FAU zeolite [65], despite the identical procedures used for the cationic exchange (except for the calcination step). Indeed, the results of Neves et al. suggest that AgZn and ZnAg-FAU (0.01 M Zn^2+^ for the exchange) have little-to-no antibacterial activity, whereas samples AgZn_0.05_ and Zn_0.05_Ag-FAU (0.5 M Zn^2+^ for the exchange) display antibacterial activity higher than the monometallic Ag sample. This discrepancy might arise from differences in the zeolitic material (FAU versus LTA), the metal content, the re-exchange process, or the antibacterial test performed. These effects will be further studied in forthcoming works.

## 4. Conclusions

The present study aims to evaluate and quantify the antibacterial effect of Ag^+^, Cu^2+^, and Zn^2+^ ions, after loading on LTA zeolite materials via cationic exchanges. The approach taken relies on careful characterisation of the exchanged materials, coupled with quantitative determination of the metal content and antimicrobial activity, using two different techniques: agar dilution and OD monitoring. For starters, the Na^+^ cations from a commercial LTA material were partially exchanged with one or two suitable solutions of metal ion salts, among Ag(NO_3_), Cu(NO_3_)_2_ and Zn(NO_3_)_2_, at different concentrations. The zeolite materials were characterised, in order to determine their metal contents (XRF, ATG), metal distributions (SEM, EDX), and structural conservations (XRD). The results indicate that all Ag^+^-exchanged zeolites samples tested (up to 11.9 wt.% Ag) display homogeneous distribution of silver over the material, whereas the zeolite structure is conserved after cationic exchange. That is not the case for Cu^2+^- and Zn^2+^-exchanged zeolite samples, for which heterogeneous metal distribution and structural loss or defects are observed at moderate loading (above 4.7 wt.% Cu and 4.6 wt.% Zn).

A range of monometallic (Ag-LTA-1 to Ag-LTA-7, Cu-LTA-4, and Zn-LTA-4) and bimetallic materials (CuAg-, AgCu-, ZnAg-, AgZn-LTA-1, Zn_0.05_Ag-, and AgZn_0.05_-LTA-2) were selected and tested towards *E. coli* growth inhibition, using two different techniques. Our results indicate that optical-density measurements and agar-dilution techniques, which both measure bacterial growth in solution after re-exchange, gave similar results in all cases (considering the concentration step and error bars). Hence, OD measurements represent a fast and reliable alternative method to the gold-standard but time-consuming, agar-dilution method, and this can be exploited for the setting of screening protocols. In the conditions tested, LTA containing 5.3 wt.% Ag inhibits *E. coli* growth at much lower concentrations (MIC50 = 0.05 mg/mL) compared to LTA containing 4.7 wt.% Cu, 4.6 wt.% Zn, and Na-LTA (MIC50 ≥ 3.5 mg/mL), which is consistent with previous studies performed with these metal ions, in solutions. A series of Ag-LTA, containing an increasing amount of Ag^+^ (0.1–11.9 wt.%), displays an increase in the antimicrobial activity that is proportional to the Ag^+^ concentration. Importantly, our results indicate that the materials remain active at Ag loading near 0.1 wt.%, thus maximising the effectiveness of the re-exchange process. Then, a bimetallic series, containing 1.2–1.5 wt.% Ag^+^ and about 0.7 wt.% to 1.4 wt.%, of either Cu^2+^ or Zn^2+^, were, also, tested, and all display superior antibacterial activity compared to monometallic Ag-LTA (considering the total amount of Ag in the suspension). However, increasing the amount of Zn^2+^ to 3.9 wt.% in the bimetallic LTA samples results in a decrease in the antibacterial properties. Surprisingly, this trend is opposite to the results previously reported in a study involving FAU-type zeolite, for which the metal content was not disclosed. Therefore, future studies will aim to evaluate the impact of the zeolite structure on the antimicrobial activity.

This work demonstrates that, by combining analytical and quantitative methods, it is possible to control the composition of mono- and bi-metal-exchanged zeolites, in order to fine-tune their antimicrobial potential, opening new ways for the development of next-generation composite zeolite-containing materials. These will have applications in multiple fields, for example, for conferring antibacterial properties to biomedical devices, such as implants and catheters, as well as various commercial products, especially, bathroom and kitchen equipment. We show that Ag^+^-exchanged LTA materials offer several advantages over their Cu^2+^ and Zn^2+^-counterparts (homogeneity, structure conservation, and antimicrobial activity) that might counterbalance the high price of silver. The addition of secondary metal ions can improve antimicrobial activity at low concentration, but their nature and content have to be carefully considered. This synergistic effect remains to be tested with other microbes (Gram-positive bacteria, yeasts, etc.), and its cytotoxicity need to be assessed before considering pharmaceutical or biomedical applications.

## Figures and Tables

**Figure 1 jfb-13-00073-f001:**
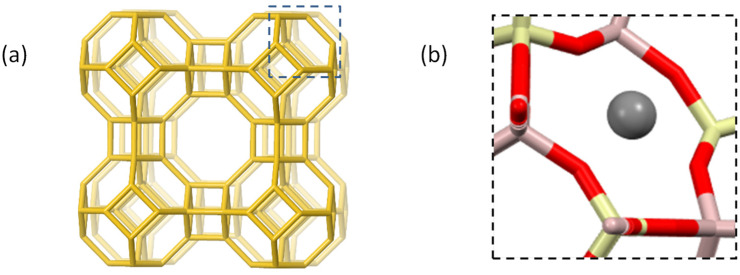
Representation of the crystal structure of the LTA zeolite Na_12_[Al_12_Si_12_O_48_]: (**a**) the unit cell displaying only T sites (Al, Si), (**b**) a six-membered ring containing a Na^+^ ion (site I); colour code Al (light pink), Si (light yellow), O (red), and Na (dark grey sphere). Adapted from the IZA database [68].

**Figure 2 jfb-13-00073-f002:**
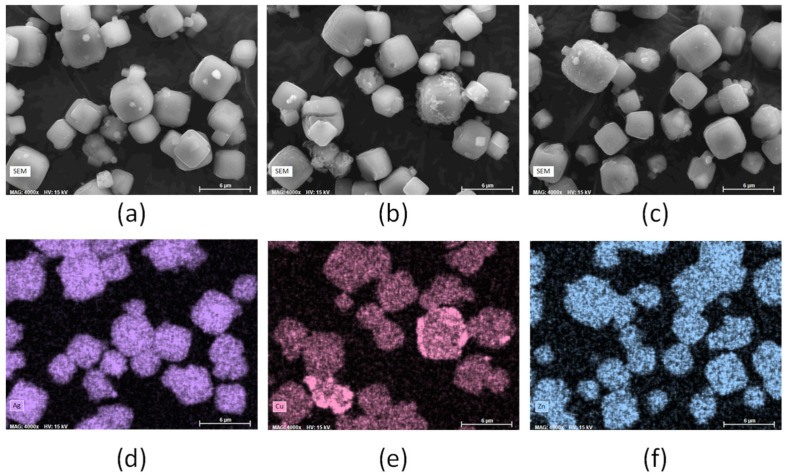
SEM pictures recorded for (**a**) Ag-LTA-5 (5.3 wt.% Ag), (**b**) Cu-LTA-4 (4.7 wt.% Cu), and (**c**) Zn-LTA-4 (4.6 wt.% Zn). EDX analyses showing Ag, Cu, and Zn distribution, respectively, in (**d**) Ag-LTA-5, (**e**) Cu-LTA-4, and (**f**) Zn-LTA-4.

**Figure 3 jfb-13-00073-f003:**
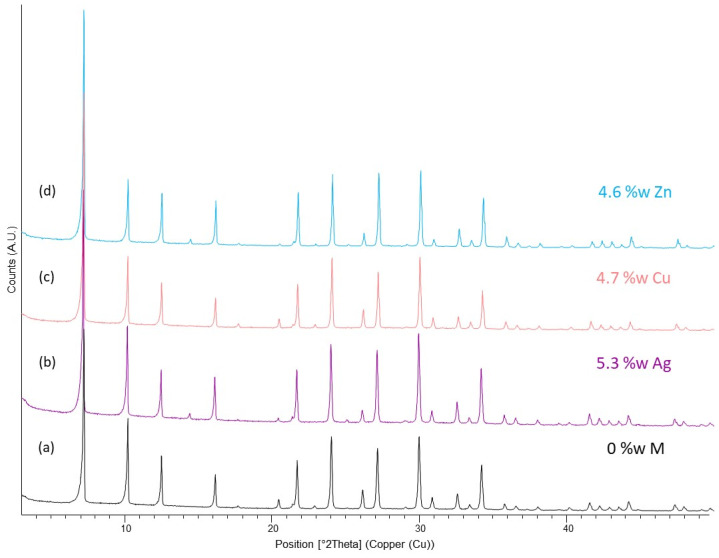
Comparison of the XRD patterns recorded for (**a**) Na-LTA, (**b**) Ag-LTA-5, (**c**) Cu-LTA-4, and (**d**) Zn-LTA-4.

**Figure 4 jfb-13-00073-f004:**
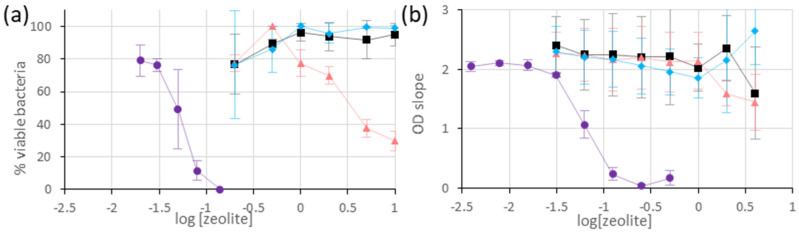
Bacterial inhibition curves for the monometallic Ag, Cu, Zn-exchanged, and non-exchanged LTA zeolites (■ 0 wt.% M, ● 5.3 wt.% Ag, ▲ 4.7 wt.% Cu, ♦ 4.6 wt.% Zn) representing either (**a**) the percentage of viable bacteria measured by agar dilution or (**b**) the optical density slope plotted versus the logarithm of the zeolite concentration (mg/mL). The values displayed are averages of two–three independent measurements, and the error bars represent standard deviations.

**Figure 5 jfb-13-00073-f005:**
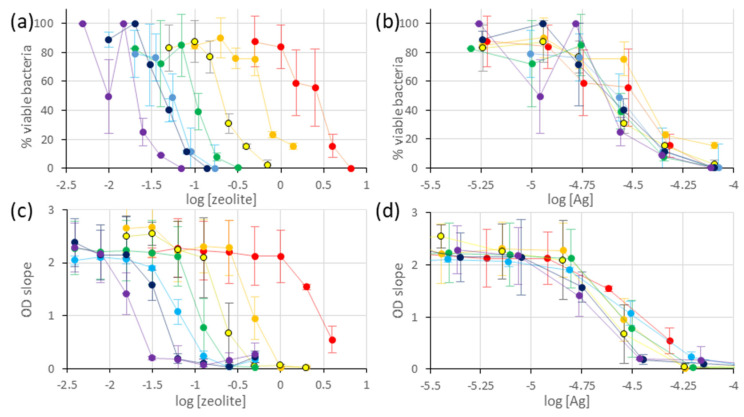
Bacterial inhibition curves for the monometallic Ag-exchanged LTA zeolites (● 0.1 wt.%, ● 0.6 wt.%, 
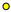
 1.2 wt.%, ● 2.7 wt.%, ● 5.3 wt.%, ● 6.1 wt.%, ● 11.9 wt.%) representing either (**a**,**b**) the percentage of viable bacteria measured by agar dilution or (**c**,**d**) optical density slope, plotted versus either (**a**,**c**) the logarithm of the zeolite concentration (mg/mL) or (**b**,**d**) the logarithm of total Ag content (M). The values displayed are averages of two–three independent measurements, and the error bars represent standard deviations.

**Figure 6 jfb-13-00073-f006:**
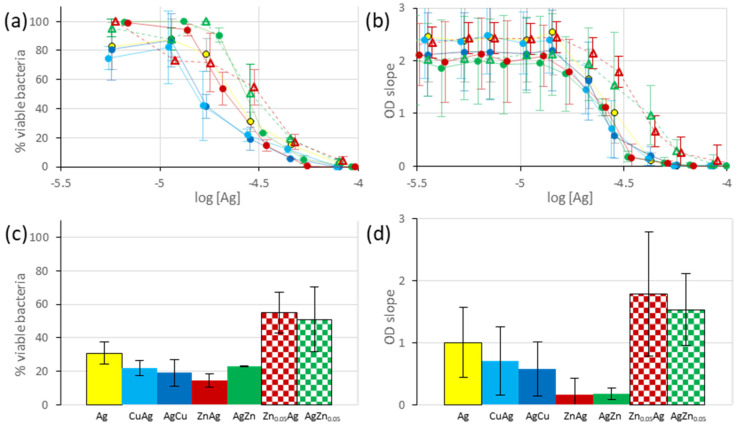
(**Top**) Bacterial inhibition curves for the bimetallic-exchanged LTA and reference monometallic Ag-LTA3 materials representing either (**a**) the percentage of viable bacteria measured by agar dilution or (**b**) the OD time slope plotted versus the logarithm of the total Ag content. (**Bottom**) Histogram representing either (**c**) the percentage of viable bacteria or (**d**) the OD time slope measured for suspension containing 0.25 mg/mL zeolite (non-corrected for Ag content). Color code: monometallic 
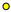
 Ag-LTA-3 (1.2 wt.% Ag), bimetallics ● CuAg-LTA-1 (0.7 wt.% Cu, 1.2 wt.% Ag) ● AgCu-LTA-1 (1.2 wt.% Ag, 0.9 wt.% Cu), ● ZnAg-LTA-1 (0.9 wt.% Zn, 1.5 wt.% Ag), ● AgZn-LTA-1 (1.4 wt.% Ag, 1.1 wt.% Zn), Δ Zn_0.05_Ag-LTA-2 (3.9 wt.% Zn, 1.3 wt.% Ag), Δ AgZn_0.05_-LTA-2 (1.2 wt.% Ag, 3.9 wt.% Zn). The values displayed are averages of two–three independent measurements, and the error bars represent standard deviations.

**Table 1 jfb-13-00073-t001:** Summary of the monometallic LTA-exchanged samples prepared, concentrations of the exchange solutions in metal ion [M^n+^], weight percentages (wt.%) of the metal in the final products, and calculated exchange rates (ER).

Name	[M^n+^] ^1^	wt.% M ^2^	Chemical Formula	ER (%)
Na-LTA	0	0	Na_13.1_K_0.2_[Si_12_Al_12_O_48_](H_2_O)_25.2_	0
Ag-LTA-1	0.001	0.1	Na_12.1_K_0.2_Ag_0.03_[Si_12_Al_12_O_48_](H_2_O)_25.0_	0.2
Ag-LTA-2	0.005	0.6	Na_12.5_K_0.2_Ag_0.1_[Si_12_Al_12_O_48_](H_2_O)_24.9_	1.0
Ag-LTA-3	0.01	1.2	Na_12.6_K_0.2_Ag_0.2_[Si_12_Al_12_O_48_](H_2_O)_24.7_	1.9
Ag-LTA-4	0.02	2.7	Na_12.2_K_0.2_Ag_0.6_[Si_12_ Al_12_O_48_](H_2_O)_25.0_	4.3
Ag-LTA-5	0.037	5.3	Na_11.5_K_0.2_Ag_1.1_[Si_12_Al_12_O_48_](H_2_O)_23.8_	8.7
Ag-LTA-6	0.05	6.1	Na_11.1_K_0.2_Ag_1.3_[Si_12_Al_12_O_48_](H_2_O)_24.8_	10.3
Ag-LTA-7	0.1	11.9	Na_9.1_K_0.2_Ag_2.7_[Si_12_Al_12_O_48_](H_2_O)_23.7_	21.9
Cu-LTA-1	0.005	0.4	Na_12.6_K_0.2_Cu_0.1_[Si_12_Al_12_O_48_](H_2_O)_25.0_	2.3
Cu-LTA-2	0.01	0.8	Na_12.8_K_0.2_Cu_0.3_[Si_12_Al_12_O_48_](H_2_O)_25.8_	4.2
Cu-LTA-3	0.025	2.5	Na_11.5_K_0.2_Cu_0.9_[Si_12_Al_12_O_48_](H_2_O)_26.1_	12.9
Cu-LTA-4	0.05	4.7	Na_9.4_K_0.2_Cu_1.6_[Si_12_Al_12_O_48_](H_2_O)_27.2_	25.5
Zn-LTA-1	0.005	0.5	Na_12.2_K_0.2_Zn_0.1_[Si_12_Al_12_O_48_](H_2_O)_24.5_	2.4
Zn-LTA-2	0.01	0.9	Na_11.8_K_0.2_Zn_0.3_[Si_12_Al_12_O_48_](H_2_O)_24.9_	4.6
Zn-LTA-3	0.025	2.5	Na_11.4_K_0.2_Zn_0.8_[Si_12_Al_12_O_48_](H_2_O)_24.4_	12.4
Zn-LTA-4	0.05	4.6	Na_10.2_K_0.2_Zn_1.6_[Si_12_Al_12_O_48_](H_2_O)_25.2_	23.1

^1^ Concentration in M, ^2^ wt.% metal was calculated from the combined results of XRF and TG measurements.

**Table 2 jfb-13-00073-t002:** Summary of the bimetallic LTA-exchanged samples prepared, concentrations of the exchange solutions in metal ion [M^n+^], weight percentages (wt.%) of the metal in the final products, and calculated exchange rates (ER).

Name	M	[M^n+^] ^1^	wt.% M ^2^	Chemical Formula	ER (%)
CuAg-LTA-1	Cu	0.01	0.7	Na_11.6_K_0.2_Ag_0.2_Cu_0.3_[Si_12_Al_12_O_48_](H_2_O)_24.7_	3.6
	Ag	0.01	1.2		1.9
AgCu-LTA-1	Ag	0.01	1.2	Na_11.7_K_0.2_Ag_0.2_Cu_0.3_[Si_12_Al_12_O_48_](H_2_O)_25.0_	2.0
	Cu	0.01	0.9		4.8
ZnAg-LTA-1	Zn	0.01	0.9	Na_11.7_K_0.2_Ag_0.3_Zn_0.3_[Si_12_Al_12_O_48_](H_2_O)_24.5_	4.6
	Ag	0.01	1.5		2.4
AgZn-LTA-1	Ag	0.01	1.4	Na_11.8_K_0.2_Ag_0.3_Zn_0.3_[Si_12_Al_12_O_48_](H_2_O)_24.3_	2.3
	Zn	0.01	1.1		5.5
Zn_0.05_Ag-LTA-2	Zn	0.05	3.9	Na_10.3_K_0.2_Ag_0.2_Zn_1.3_[Si_12_Al_12_O_48_](H_2_O)_24.9_	19.6
	Ag	0.01	1.3		2.0
AgZn_0.05_-LTA-2	Ag	0.01	1.2	Na_10.1_K_0.2_Ag_0.2_Zn_1.3_[Si_12_Al_12_O_48_](H_2_O)_24.5_	1.9
	Zn	0.05	3.9		20.3

^1^ Concentration in M, ^2^ wt.% metal was calculated from the combined results of XRF and TG measurements.

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
