# Peer review of "Rational Design and Characterisation of Novel Mono- and Bimetallic Antibacterial Linde Type A Zeolite Materials"

_jfb, 2022, doi:10.3390/jfb13020073_

Round 1
Reviewer 1 Report
The manuscript presents interesting studies on the preparation of monometallic and bimetallic zeolites and their ability to inhibit bacterial growth. The information provided by the authors is valuable and scientifically interesting. The manuscript is written in a clear style.
- The use of abbreviations in the title (LTA zeolite) is unacceptable. Therefore, the title of manuscript should be corrected.
- The information in lines 208-210, 317-319, 324-328, etc. of the Results and Discussion section should be moved to the Materials and Methods section.
- The results of studies on the ability of monometallic and bimetallic zeolites to inhibit the growth of gram-positive bacteria would also be helpful.
Author Response
Reviewer 1
Comment:
The manuscript presents interesting studies on the preparation of monometallic and bimetallic zeolites and their ability to inhibit bacterial growth. The information provided by the authors is valuable and scientifically interesting. The manuscript is written in a clear style. The use of abbreviations in the title (LTA zeolite) is unacceptable. Therefore, the title of manuscript should be corrected.
Response :
We thank the reviewer for the supportive comments on the manuscript content and writing. The reviewer rightfully pointed out the abbreviations “LTA” in the title as unacceptable. Therefore, we modified the title to display the full name of the zeolite structural type “Linde Type A”.
Comment:
The information in lines 208-210, 317-319, 324-328, etc. of the Results and Discussion section should be moved to the Materials and Methods section.
Response:
According to the reviewer comment, the information in lines 208-210, 317-319 have been deleted since they are already mentioned in the Materials and Methods in more details. The information in lines 324-328 has been moved to the materials and methods section and minor changes were included in the text such as the mention of Figure 4a still appears in the results and discussion. It is not clear what the reviewer suggest by “etc.”, therefore we applied similar deletion at lines 384-385 and the lines 390-392 were displaced to the material and methods section. Please note that the latter line numbers correspond to those of the corrected manuscript with track changes.
Comment:
The results of studies on the ability of monometallic and bimetallic zeolites to inhibit the growth of gram-positive bacteria would also be helpful.
Response:
We agree with the reviewer comment that the work on gram-positive bacteria could complement this study and we will carry out these experiments in forthcoming works.
Reviewer 2 Report
The manuscript entitled “Quantifying the antimicrobial activity of LTA zeolite materials exchanged with Ag+, Cu2+, and/or Zn2+ metal-ions” was revised. In this work they study the properties of Linde Type A zeolites exchanged with Ag+, Cu2+, and/or Zn2+ metal-ions. The materials is characterized using SEM-EDX (metal distribution), XRD (follow the structural conservation), XRF, ATG) (for the metal content).
It is an important research topic. This submission is an interest but needs more English revision. Should supports by references especially in the introduction section. I therefore recommend being considered for publication after minor revision.
- The overall writing should be improved. Refine the language of the manuscript and check all grammar errors and mistakes.
- Author should precise the introduction part and cite it with recent references relate to study.
- The equipment used should be provided with the manufacture country.
- The conclusion section should be precise with more information.
Author Response
Comment:
The manuscript entitled “Quantifying the antimicrobial activity of LTA zeolite materials exchanged with Ag , Cu , and/or Zn metal-ions” was revised. In this work they study the properties of Linde Type A zeolites exchanged with Ag , Cu , and/or Zn metal-ions. The materials is characterized using SEM-EDX (metal distribution), XRD (follow the structural conservation), XRF, ATG) (for the metal content).
It is an important research topic. This submission is an interest but needs more English revision. Should supports by references especially in the introduction section. I therefore recommend being considered for publication after minor revision.
Response:
We thank the reviewer for his positive comments on the manuscript and will address his remarks in the corrected version of the manuscript.
Comment:
The overall writing should be improved. Refine the language of the manuscript and check all grammar errors and mistakes.
Response:
Following on the reviewer comment, the manuscript was checked for grammar error and mistakes. Grammar mistakes at lines 24, 26, 42, 62, 64, 65, 77, 81, 88, 100, 130, 164, 168, 204, 219, 227, 245, 247, 248, 249, 270, 302, 306, 350, 359, 373, 376, 401, 403, 407, 412, 431, 442, 452, 476, 482, 483, 496, 497, 516 were corrected.
Comment:
Author should precise the introduction part and cite it with recent references relate to study.
Response:
We agree with the reviewer that the introduction part should cite recent references related to the study. In the manuscript, the section between lines 56-88 describes the previous works on the topic of antimicrobial zeolites. In this section 26 references on Ag, Cu, Zn exchanged zeolites used in this context are cited, moreover 9 references on composite materials and 9 references on coatings are also cited. Then 2 reviews particularly relevant to this topic are cited. This work is compared to six other references involving LTA materials and a detailed account of the results obtained by Neves and co-worker is given. The articles cited were all published in the last 25 years and we believe it represents an exhaustive list of previous work on this topic. We would gladly add any additional references that we have missed.
Comment:
The equipment used should be provided with the manufacture country.
Response:
The reviewer is right and the manufacture country of the equipment was added to the manuscript at lines 128, 132, 146, 148 and 149.
Comment:
The conclusion section should be precise with more information.
Response:
We thank the reviewer for this comment. Additional information were added at lines 479, 481 and 497-499 in order to improve the overall quality and precision of the conclusions on the nature of metal-ions, techniques used to assess the antimicrobial activity and concerning the results of this activity on the different materials.
Reviewer 3 Report
The manuscript entitled "Quantifying the antimicrobial activity of LTA zeolite materials exchanged with Ag+, Cu2+, and / or Zn2+ metal-ions", reports a study on the antimicrobial activity of zeolites, LTA, exchanged with Ag+, Cu2+ and Zn2+ metals. Initially a single exchange was carried out with only one metal at a time (silver, copper or zinc ion). Subsequently a double exchange was carried out, that is, in addition to the Ag+ ion, the copper or zinc ion was also exchanged. In this last case, the exchange order of the two metals was also considered. Several samples of exchanged zeolite were obtained and tests were carried out on them.
The work is interesting although it could be improved especially in its exposure. A more marked distinction between methods and results is desirable. Therefore the manuscript can be considered for its publication only after a minor revision. Here are some tips for authors.
*) The title could possibly be improved, it is not very clear. It does not represent well what has been done ".... Ag+, Cu2+, and / or Zn2+".
*) (Lines 81-84). The manuscript contains the following sentence "… we selected a common zeolite with high Si / Al ratio (= 1), the Linde type A (LTA) (Figure 1), thus allowing for metal-exchange at high rate whilst limiting the formation of undesired metal aggregates. ”This statement should be better explained. Why does a high Si / Al ratio allow for a higher exchange rate? What are the references that state this? However, a zeolite with a high Si/Al ratio should have a low amount of exchangeable ion.
*) Paragraph 3.1. The first part appears to be a repetition of paragraph 2.2. Perhaps it would be better to integrate this first part in paragraph 2.2. The same is true for paragraph 3.2.
*) Line 207. The following sentence is reported in the manuscript “… one step for monometallic exchanged samples and in two successive steps for bimetallic exchanged samples (Figure 1)”. The reference to figure 1 is not clear.
*) 4.Conclusions instead of 5.Conclusions
Author Response
Comment:
The manuscript entitled "Quantifying the antimicrobial activity of LTA zeolite materials exchanged with Ag , Cu , and / or Zn metal-ions", reports a study on the antimicrobial activity of zeolites, LTA, exchanged with Ag , Cu and Zn metals. Initially a single exchange was carried out with only one metal at a time (silver, copper or zinc ion). Subsequently a double exchange was carried out, that is, in addition to the Ag+ ion, the copper or zinc ion was also exchanged. In this last case, the exchange order of the two metals was also considered. Several samples of exchanged zeolite were obtained and tests were carried out on them. The work is interesting although it could be improved especially in its exposure. A more marked distinction between methods and results is desirable. Therefore the manuscript can be considered for its publication only after a minor revision. Here are some tips for authors.
Response:
We agree with the reviewer and thank him/her for the positive feedbacks. Based on the comments from reviewer 1, four sections of text have been moved from the “results and discussions” section to the “materials and methods” section and we believe these changes provide the manuscript with a better distinction between methods and results.
Comment:
*) The title could possibly be improved, it is not very clear. It does not represent well what has been done ".... Ag , Cu , and / or Zn ".
According to the reviewer comment, the title has been changed for “Rational design and characterisation of novel mono- and bimetallic antibacterial Linde Type A zeolite materials”.
Comment:
*) (Lines 81-84). The manuscript contains the following sentence "… we selected a common zeolite with high Si / Al ratio (= 1), the Linde type A (LTA) (Figure 1), thus allowing for metal-exchange at high rate whilst limiting the formation of undesired metal aggregates. ”This statement should be better explained. Why does a high Si / Al ratio allow for a higher exchange rate? What are the references that state this? However, a zeolite with a high Si/Al ratio should have a low amount of exchangeable ion.
Response:
We thank the reviewer for pointing out this mistake. The text was corrected to mention low Si/Al ratio (instead of high Si/Al). In order to better explain the statement, the mention “associated with a high content of negatively charged aluminates” was added to the text.
Comment:
*) Paragraph 3.1. The first part appears to be a repetition of paragraph 2.2. Perhaps it would be better to integrate this first part in paragraph 2.2. The same is true for paragraph 3.2.
Response:
The paragraph 2.2 was considerably shortened as per the reviewer comment and the information was merged in the materials and methods section such as repetitions are avoided. The first part of paragraph 3.2 was equally moved to the Materials and methods section 2.2 and the text was adapted accordingly. As a consequence, lines 134-141 were also moved from section 2.1 to 2.2 such as the determination of the metal content is fully described in section 2.2.
Comment:
*) Line 207. The following sentence is reported in the manuscript “… one step for monometallic exchanged samples and in two successive steps for bimetallic exchanged samples (Figure 1)”.
The reference to figure 1 is not clear.
Response:
We agree with the reviewer that the reference to figure 1 is not clear in this case. The first sentence of the paragraph was divided in two and the mention “from the sodium-containing LTA (Na-LTA) starting material” was added to the text, thus describing the content of figure 1 in the text.
Comment:
*) 4.Conclusions instead of 5.Conclusions
Response:
Indeed, the reviewer is right and the section numbering was corrected according to the reviewer feedbacks.